# New Axes of Interaction in Circ_0079593/miR-516b-5p Network in Melanoma Metastasis Cell Lines

**DOI:** 10.3390/genes15121647

**Published:** 2024-12-21

**Authors:** Elisa De Tomi, Elisa Orlandi, Francesca Belpinati, Cristina Patuzzo, Elisabetta Trabetti, Macarena Gomez-Lira, Giovanni Malerba

**Affiliations:** 1Department of Neurosciences, Biomedicine, and Movement Sciences, University of Verona, 37134 Verona, Italy; elisa.detomi@univr.it (E.D.T.); elisa.orlandi@univr.it (E.O.); francesca.belpinati@univr.it (F.B.); cristina.patuzzo@univr.it (C.P.); elisabetta.trabetti@univr.it (E.T.); 2Department of Surgery, Dentistry, Paediatrics and Gynaecology, University of Verona, 37134 Verona, Italy; giovanni.malerba@univr.it

**Keywords:** circularRNAs, microRNAs, melanoma, miR-516b-5p, metastasis, CHAF1B, MCAM

## Abstract

Background/Objectives: microRNAs (miRNAs) and circular RNA (circRNAs) show a close interconnection in the control of fundamental functions, such as cell proliferation and tumor development. A full understanding of this complex and interconnected network is essential for better understanding the mechanisms underlying cancer progression. Hsa_circ_0079593 is a circRNA highly expressed in melanoma and is associated with increased metastasis and progression of malignancy, whereas miR516b-5p is a microRNA whose expression is lower in several tumor types, including melanoma; its overexpression inhibits cell proliferation, migration, and invasion. In this study, we tested whether circ_0079593 is involved in the progression of melanoma aggressiveness by regulating CHAF1B and MCAM via the inhibition of miR-516b-5p. Methods: We first verified the expression of the key components in both healthy melanocyte lines and melanoma metastases, subsequently using in vitro assays such as scratch tests, Western blot, qRT-PCR, and dual luciferase report assay; we verified their interconnected regulatory effect. Results: Our results showed that circ_0079593-miR516b-5p interactions are involved in the increase in the migration of metastasis melanoma cells by exploiting their binding to MCAM and CHAF1B mRNAs. Conclusions: This study provides two other regulatory networks in which circ_0079593 may exert its oncogenic function by increasing the speed of movement of metastatic cells through the sponge of miR-516b-5p, which cannot regulate MCAM and CHAF1B expression.

## 1. Introduction

In recent years, there has been significant focus on the role of non-coding RNAs (ncRNAs) and their involvement in cancer development and progression [1,2,3,4,5]. ncRNAs, such as circular RNAs (circRNAs) and microRNAs (miRNAs), although not coding for proteins, play a crucial role in regulating gene expression and modulating various cellular processes [6,7,8,9,10].

MicroRNAs are short RNA molecules that negatively regulate post-transcriptional gene expression. They affect protein production by binding to complementary sequences on their target messenger RNAs (mRNAs), preventing their translation, or promoting their degradation [11]. Conversely, circRNAs are a particular class of single-stranded ncRNAs with a covalently closed circular structure that lends them a high stability [12]. One of their many abilities is to function as competitive endogenous RNAs (ceRNAs) by acting as cellular sponges. CircRNAs, sequestering miRNAs complementary to their sequences, can prevent the interaction with their target mRNAs and, indirectly, modulate the gene expression [11,12,13,14].

This suggests that interactions between these regulatory elements are interconnected. Aberrant expression of any component of the ncRNA network can destabilize the entire regulatory system, contributing to cancer development and progression [15,16,17]. Several studies have shown that the expression of these regulatory molecules changes between cell and tissue types, performing both oncogenic and tumor-suppressor effects depending on the context [18,19]. This duality reflects the complexity of the regulatory interactions of ncRNAs that, as reported in numerous studies, influence cancer cell proliferation, invasion, and metastasis [19].

Cutaneous melanoma is a highly aggressive form of cancer that tends to metastasize at an early clinical stage [1,20]. Despite the ongoing advancements in cancer therapies, this type of skin cancer is characterized by an intrinsic resistance to treatments. It is necessary for researchers to better clarify the mechanisms associated with the progression of melanoma and the development of metastasis, to identify new therapeutic strategies.

In this context, the dysregulation of circRNAs, identified as a critical factor in melanoma progression, may also play a role in the development of treatment resistance [14,21]. To date, the characterization of circRNAs in melanoma is still in its nascent stage. An increasing number of new circRNAs are reported to contribute to melanoma development [22,23]. Recent studies report that circ_0079593 is overexpressed in melanoma, and its inhibition has been shown to suppress melanoma progression, while its presence has been associated with increased proliferation, metastasis, glucose metabolism, and inhibition of apoptosis through the regulation of the miR-433/EGFR (Epidermal growth factor receptor), miR-573/ABHD2 (Abhydrolase Domain Containing 2, Acylglycerol Lipase), and miR-516b/GRM3 (Glutamate Metabotropic Receptor 3) axes [24,25,26]. These findings suggest the involvement of other regulatory networks.

This study aimed to determine whether the circ_0079593, sponging miR-516b-5p, could modify the regulation of two melanoma-associated genes, Melanoma Cell Adhesion Molecule (MCAM) and Chromatin Assembly Factor 1 Subunit B (CHAF1B) in two different types of melanoma metastasis cell lines.

## 2. Results

### 2.1. Expression Pattern of Circ_0079593 in Melanoma Cell Lines and Primary Melanocytic Cells and Its Role in Melanoma Progression

We validated the expression pattern of circ_0079593 and the corresponding linear mRNA (IGF2BP3, Insulin Like Growth Factor 2 MRNA Binding Protein 3) in melanoma cell lines and primary melanocytic cells with and without RNase R enrichment. As described in the literature, qRT-PCR (quantitative real-time PCR) showed a progressive increase in the expression of circ_0079593 in relation to increased tumor aggressiveness in different types of melanoma lines (Figure 1). As expected, we observed a very low circRNA expression level in Normal Human Epidermal Melanocytes (NHEM), while in A375 (commercial cutaneous melanoma) and in LM-16 and LM-36 (metastatic cell lines), circRNA resulted overexpressed, particularly in LM-36 (Figure 1a). A similar pattern progression was detected in the expression of the parental gene (IGF2BP3), with a reversal between the lymph node metastatic cell line (LM-16) and the skin metastatic cell line (LM-36) (Figure 1b). In both cases, RNase R endonuclease significantly reduced linear mRNA levels, demonstrating the intrinsic stability of circRNA and the accuracy of the designed primers.

To further investigate the role of circ_0079593 in the progression of melanoma, a wound healing assay was used to assess the cell migration capacity. To do this, LM-16 and LM-36 cells were transfected with circ_0079593 siRNA (silencing RNA) to inhibit endogenous circ_0079593 expression and the maximum transfection efficiency was determined using qRT-PCR at different time points (Figure 1c,d). Figure 1e–h show that siRNA transfection inhibited cell migration potential in both cell lines in different ways. In particular, LM-36 cells (Figure 1e,g) showed a progressive reduction in cell migration starting from 4 h after transfection, with a significant difference compared to the negative control after 16 and 24 h. In contrast, LM-16 cells (nodular metastasis) showed high cell migration potential, with complete wound closure 16 h after transfection and a significant difference from the negative control after only 4 h (Figure 1f,h).

### 2.2. The Potential of miR-516b-5p as a Target for Circ_0079593 and Its Biological Effects on Melanoma Metastasis

To identify a potential regulatory axis exploited by circ_0079593 in melanoma progression, we tested target miRNAs predicted by the CircInteractome (https://circinteractome.irp.nia.nih.gov accessed on 13 January 2023) (Appendix A). After an extensive literature review, we decided to validate the results of Zhao et al. (2021) [25] investigating miR-516b-5p in melanoma metastatic lines.

Using RT-PCR, we analyzed the miRNA expression profiles of A375, LM-36, and LM-16 cell lines compared to those of healthy NHEM cells (Figure 2a). In contrast to the findings of Zhao et al., no detectable levels of miR-516b-5p were present in melanocytes and in the LM-16 metastatic cell line. The skin metastasis line LM-36, on the other hand, showed high expression.

Given the difference in both cell lines, we next investigated the biological effects of miR-516b-5p in melanoma cells using a wound healing assay. Taking advantage of the expression profile of this microRNA, we overexpressed and inhibited miR-516b-5p in LM-16 and LM-36 cell lines, respectively. Transfection efficiency was evaluated using qRT-PCR (Figure 2b). Images and graphs highlighting the effect of transfection on the migration potential of both cell lines are shown in Figure 2d–g. Specifically, the wound healing assay showed that miR-516b-5p overexpression inhibited tumor migration in LM-16 cells (Figure 2d,f), and miR-516b-5p inhibition enhanced cell migration in LM-36 cells (Figure 2e,g).

To confirm whether circ_0079593 can sponge miR-516b-5p, we used a dual luciferase reporter assay on A375 and LM-16 cells. The circ_0079593 sequence was cloned into a luciferase-containing plasmid, and a mutant plasmid was created by site-directed mutagenesis by substituting two nucleotides to interfere with the miR-516b-5p binding site. The results showed that overexpression of miR-516b-5p significantly reduced luciferase activity in the presence of the wild-type circ0079593-pGL3 plasmid. No change was detected in the presence of the corresponding mutated plasmid (circ0079593-pGL3-MUT) or empty control pGL3 promoter (Figure 2h). This suggests that circ_00079593 can sponge miR-516b-5p.

To further confirm this interaction, siRNA inhibition of circ_00795993 showed a significant increase in endogenous miR-516b-5p in all cell lines at 12 h post-transfection (Figure 2i). This was even observed in the LM-16 cell line, which showed no detectable miR-516b-5p levels after qRT-PCR amplification in basal condition (Figure 2a).

### 2.3. MiR-516b-5p Targets CHAF1B and MCAM in Melanoma Metastasis Cell Lines

To identify the downstream target mRNAs of miR-516b-5p, we used the multiMiR R package (http://multimir.org) to interrogate several databases that predict miRNA-mRNA interactions. Among the candidates obtained, MCAM and CHAF1B were found to be the most significantly overexpressed genes in melanoma metastasis cells, along with no miR-516b-5p levels (LM-16) and with high miR-516b-5p levels in LM-36 (Figure 3a).

To determine whether miR-516b-5p can bind directly to CHAF1B and MCAM mRNAs and regulate them via their 3’UTR, a luciferase reporter assay was performed in A375 and LM-16 cells. This assay revealed a significant reduction in luciferase activity in wild-type plasmids (PGL3 + 3′UTR_CHAF1B, PGL3 + 3′UTR_MCAM) in the presence of miR-516b-5p overexpression. In contrast, it showed no significant differences in the presence of plasmids containing the mutated binding site (PGL3 + 3′UTR_CHAF1B_MUT, PGL3 + 3′UTR_MCAM_MUT) (Figure 3b), indicating that CHAF1B and MCAM are the direct targets of this miRNA.

Further in vitro experiments showed that the overexpression of miR-516b-5p in LM-16 cells, which basally do not present the miR, revealed a non-significant reduction in CHAF1B and MCAM mRNAs expression at 24 h (Figure 3c), indicating a poor effect on mRNA stability. The inhibition of miR-516b-5p in LM-36 leads to the significant upregulation of MCAM and CHAF1B (Figure 3e), indicating that there is a link between miR-516b-5p and the two genes in LM-36.

Western blot analysis showed that the overexpression of miR-516b-5p significantly reduced MCAM and CHAF1B protein levels (Figure 3d—LM-16), and the inhibition of miR-516b-5p led to a small but significant increase in MCAM and CHAF1B protein levels compared to the control (Figure 3f—LM-36), suggesting that miR-516-5p is involved in the protein translation of both genes.

## 3. Discussion

Malignant melanoma is a form of skin cancer difficult to treat if not detected early and with high metastatic potential [27]. MicroRNAs and circRNAs are two classes of ncRNAs, that have emerged as important key regulators of gene expression in melanoma and cancer in general [1,2,28]. It has been shown that the respective knockdown of circ_0023988, circ_0008157, and circ_0030388 enhances the proliferation and invasion of low-metastatic melanoma cells, suggesting the role of these circRNAs as potential tumor suppressors [3]. In contrast, the overexpression of circ_0025039 in melanoma tissues can promote growth, cell invasion, and glucose metabolism by acting as a sponge for miR-198, suggesting the involvement of specific circRNA-miRNA networks in tumor progression processes [29]. Despite this, the mechanism by which these classes of ncRNAs influence melanoma progression remains largely unknown, and a better understanding of the regulatory axes mediated by circRNAs and microRNAs could help both to identify new therapeutic targets for the treatment of melanoma and to understand the tumor progression that underlies the aggressiveness of this cancer [30,31].

In this study, we analyzed the role of the regulatory axis composed of circ_0079593 and miR-516b-5p in two types of melanoma metastasis cell lines (LM-36 and LM-16), with a focus on their interaction with the mRNAs of MCAM and CHAF1B, both at high expression levels associated with an aggressive phenotype [32,33,34]. In particular, CHAF1B is a chromatin assembly factor that regulates DNA replication and repair, and high levels of this protein induce a resistance to radiotherapy. MCAM, instead, is a cell adhesion molecule involved in cell migration and invasion. This protein is a plasma biomarker in patients with melanoma, and high levels are associated with the development of recurrence [33,35,36,37]. The two cell lines used are LM-36, a line derived from melanoma skin metastasis biopsy and LM-16, a melanoma metastasis line extracted from lymph node biopsy [38].

At first, we assessed the relative expression of all the components of the circ_0079593/miR-516b-5p regulatory axis. Following the trend of other studies, we confirmed the high expression of circ_0079593 in melanoma cells, and that this overexpression increases with metastasis progression compared to cutaneous melanoma (A375) [24,26]. Indeed, LM-16, a lymph node metastasis, showed a higher expression of the circRNA with respect to LM-36 a skin metastasis. As demonstrated by Lu and Li [26] on melanoma cells, through the dual luciferase report assay, we confirmed the direct interaction of circ_0079593 with miR-516b-5p in LM-16 and A375 cell lines. However, in contrast to previous studies that indicated that miR-516b-5p is mainly a tumor-suppressor miR and is therefore low expressed in tumor cells [39,40], we observed that miR-516b-5p was expressed at extremely lowly levels in the primary healthy melanocyte line (NEHM) and was highly expressed in cutaneous metastatic melanoma (LM-36). Nevertheless, consistent with the literature [41], this miRNA was almost completely absent in the lymph node metastatic melanoma line (LM-16). This difference could be explained by the high expression of circ_0079593 in LM-16, which could act as a molecular “sponge”, sequestering miR-516b-5p and reducing its availability for regulatory function. In support of this hypothesis, in the context of melanoma, it has also been reported that miR-516b-5p expression is lower in aggressive than in less aggressive primary tumors and that overexpression reduces metastatic invasion [41]. Subsequently, we confirmed, in agreement with data published by Lu and Li [26] and Zhao and colleagues [25] on A375 and SK-MEL-2 cell lines, that circ_0079593 silencing reduced tumor cell migration (wound scratch assay) and increased the endogenous availability of miR-516b-5p in melanoma metastatic cell lines, making it detectable even in LM-16 cells.

Subsequently, we identified MCAM and CHAF1B as potential targets of circ_0079593/miR-516b-5p regulatory axis, and their different gene expression in LM-16 and LM-36 cells appeared to be influenced by the expression levels of the examined ncRNAs. We confirmed that MCAM and CHAF1B were overexpressed compared to the healthy melanocyte cell line (NHEM) in both the commercial cutaneous melanoma cell line (A375) and short-term cutaneous and nodal metastatic melanoma cell lines (LM-36 and LM-16, respectively). Additionally, we observed that the expression of these genes was extremely elevated in LM-16, whereas they were overexpressed in LM-36 compared to healthy cells but much less expressed in relation to LM-16.

Through the overexpression and inhibition of miR-516b-5p in LM-16 and LM-36 cells, respectively, we demonstrate the participation of miR-516b-5p in the regulation of MCAM and CHAF1B genes and proteins. In addition, the dual luciferase report assay demonstrated the direct interaction between miR-516b-5p and its binding site in the 3’UTR region of MCAM and CHAF1B, demonstrating that MCAM and CHAF1B are functional targets of miR-516b-5p.

In terms of the profile expression of ncRNAs in LM-36 and LM-16 melanoma metastatic cells, we can hypothesize that circ_0079593 might act as a pro-cancer factor, particularly in the lymph node metastatic cell line LM-16. The biological significance of the high expression of miR-516b-5p in LM-36, despite the high endogenous presence of circ_0079593, requires further investigation. Although these findings provide valuable insights, they also highlight the complexity of the landscape in melanoma. Specifically, the genetic diversity among melanoma, while posing challenges, offers opportunities for developing new therapeutic strategies using ncRNAs.

In conclusion, our results suggest that circ_0079593 can promote the migration of metastatic melanoma cells, increasing aggressiveness, through the regulation of the miR-516b-5p/MCAM and miR-516b-5p/CHAF1B axes. Further studies are needed to fully understand the mechanism of this regulation and its potential implications in melanoma progression and treatment. In this context, our study not only adds to the current knowledge but also opens up new avenues for understanding the mechanisms underlying melanoma and the potential development of targeted therapies.

## 4. Materials and Methods

### 4.1. Cell Culture

A primary Normal Human Epidermal Melanocytes line (NHEM, C-12400, PromoCell, Heidelberg, Germany) and three melanoma cell lines, A375 (CRL-1619, ATCC, Manassas, VA, USA), LM-16 (4023M), and LM-36 (26414M), were utilized. LM-16 and LM-36 cell lines, derived from melanoma metastasis biopsies, specifically from the lymph node (LM-16) and skin (LM-36), were provided by Dr. Monica Rodolfo from the Istituto Nazionale Tumori di Milano [38].

All cell lines were cultured in RPMI 1640 medium supplemented with 10% heat-inactivated FBS, 1% L-glutamine (2 mM), and 2% penicillin-streptomycin (5000 IU/mL and 5000 µg/mL, respectively), all purchased from Gibco (Thermo Fisher Scientific, Waltham, MA, USA). The cultures were incubated at 37 °C in a humidified atmosphere containing 5% CO_2_.

### 4.2. Transfection Procedures for miRNA and siRNA

To perform miRNAs transfections, 50 nM of miR-516b-5p mimic and inhibitor (Cat. No. MCH02871, Cat. No. MIH02871) or matched negative controls (miR-NC Cat. No. MCH000000, anti-NC Cat. No. MIH00000) from Applied Biological Materials Inc. (Richmond, BC, Canada) were transfected into LM-16 and LM-36 cell lines, respectively, using Lipofectamine 3000 in Gibco™ OptiMEM medium (Thermo Fisher Scientific) following the manufacturer’s instructions. RNA extraction was performed at 24 h, and protein extraction was performed at 72 h for inhibitor and at 96 h mimic transfection, respectively.

To inhibit the circ_0079593, 35pmol of a custom pool of small interference RNA (siRNAs), designed to target the junction between exon 10 and 9 of circ_0079593 (IGF2BP3 gene), or its negative control (Cat No. A06001), all purchased from GenePharma (Pudong New Area, Shanghai, China), were transfected into the A375, LM-36, and LM-16 cell lines, as reported previously. The siRNA pool sequences targeting the circ_0079593 were 5’-ACCCGCAGUUUGAGAUUGCTT-3’ and 5’-AGUUUGAGAUUGCAGGAAUTT-3’.

After transfection, cells were collected for RNA extraction and analysis at 4, 8, 12, and 16 h.

### 4.3. Relative Quantification of miRNA mRNA and CircRNA with Real-Time PCR

Total RNA was extracted from the cells using TRIzol™ reagent (Thermo Fisher Scientific). For mRNA and circRNA analyses, cDNAs were synthesized using the SensiFast cDNA Synthesis Kit (Meridian Bioscience, Cincinnati, OH, USA), according to the manufacturer’s instructions. Before cDNA synthesis for circRNA analysis, a circular RNA enrichment step was performed using RNAase R (Applied Biological Materials Inc., Cat. No. E049). qRT-PCR was conducted using the SensiFAST™ SYBR No-ROX Kit (Meridian Bioscience). Normalization was performed using the TATA box protein (TBP) gene expression. The primers used for amplification are listed in Table 1.

For microRNA analysis, cDNAs were synthesized using TaqMan Advanced miRNA cDNA Synthesis Kit, and qRT-PCR reactions were performed using the TaqMan™ Fast Advanced Master Mix Kit using specific probes of miR-516b-5p (478979_mir, Thermo Fisher Scientific), and miR-191-5p (477952_mir, Thermo Fisher Scientific), following the manufacturer’s guidelines. miR-191-5p was used as a reference for miRNA normalization.

Each measurement was performed in triplicates and in at least three different experiments. The ΔΔCt method was used to calculate the relative fold-change [42].

### 4.4. Protein Isolation and Western Blot

Cells were lysed with RIPA buffer supplemented with protease inhibitors cocktail (Roche), and the concentrations were determined using the Bradford assay (Thermo Fisher Scientific). Equal amounts of protein were separated by 10% SDS-PAGE and electroblotted onto PVDF membranes (Thermo Fisher Scientific). Membranes were saturated with 5% non-fat milk in TBST solution, hybridized with specific primary antibodies, and incubated with anti-rabbit HRP-conjugated secondary antibody (1:6000, Cell Signaling Technology, Danvers, MA, USA). All primary antibodies, purchased from Wuhan Fine Biotech Co., Ltd., (Wuhan, China) were diluted in 5% non-fat milk in TBST solution at the following concentrations: anti-β-actin (1:2000), anti-TBP (1:2000), anti-CHAF1B (1:1000), and anti-CD146/MCAM (1:1000). Bound antibody signals were detected using a WesternBright ECL kit (Advansta Inc., San Jose, CA, USA) and images were acquired using an Azure C300 Processing machine (Azure Biosystems, Dublin, CA, USA). ImageJ 1.53 software was used for densitometric analysis and protein expression was normalized to β-actin and anti-TBP. Each measurement was performed in triplicate with a minimum of three independent experiments.

### 4.5. Scratch Wound Healing Test

The migration of LM-16 and LM-36 cells was performed using the scratch test method. Cells were seeded in a 12-well plate upon reaching 85–90% confluence, the cell monolayer was scratched with a sterile pipette tip, and the detached cells were washed with PBS. Subsequently, RPMI medium containing 0.5% fetal bovine serum (FBS) was added. Immediately following scratching, the cells were transfected with the miR-516b-5p mimic, miR-516b-5p inhibitor, and siRNA of circ_0079593 or matched negative control, as previously described. Images were captured at 4, 8, 16, 24, and 32 h post scratch using an inverted microscope (Axio Vert A1, Zeiss, Oberkochen, Germany) and analyzed using ImageJ.

### 4.6. Plasmid Construction and Dual Luciferase Reporter Assay

To generate the wild-type plasmids pGL3_3’UTR_CHAF1B_WT, pGL3_3’UTR_MCAM_WT and pGL3_circ_0079593_WT, a portion of the 3’UTR region of the CHAF1B and MCAM genes, containing the miR-516b-5p binding site, and the entire sequence of circ_0079593, were subcloned into the pGL3 promoter vector (Promega, Madison, WI, USA) downstream of the firefly luciferase gene. Plasmid sequences were confirmed by restriction digestion and Sanger sequencing (BMR Genomics, Padova, Italy).

The mutagenesis to remove the miR-516b-5p binding site from the prior plasmid was performed using the QuikChange Site Directed Mutagenesis Kit (Stratagene, Agilent Technologies, Santa Clara, CA, USA), according to the manufacturer’s instructions. The sequences of the wild-type and mutated plasmids (pGL3_3’UTR_CHAF1B_MUT, pGL3_3’UTR_MCAM_MUT, and pGL3_circ_0079593_MUT) were confirmed by enzymatic digestion and Sanger sequencing. All primers used for amplification and mutagenesis are listed in Table 2.

For the dual luciferase reporter assay, LM-16 and A375 cells were transiently co-transfected with 1 µg of empty, wild-type, or mutagenized pGL3 promoter vector; 50 nM miR-516b-5p mimic or miR-NC; and Renilla luciferase control plasmid vector (pRL null, Promega). After 24 h, Firefly and Renilla luciferase activities were measured using the Dual-Luciferase Assay Reporter System (Promega), following the manufacturer’s protocol. Each measurement was performed in duplicate with a minimum of three independent experiments. Luciferase activity was calculated after subtracting the activity levels of the cells alone.

### 4.7. Statistical Analysis

All data were analyzed using the GraphPad Prism 7.03 statistical program (GraphPad Software Inc., Boston, MA, USA) using an unpaired, two-tailed Student t-test. Statistical significance was set at *p* < 0.05 and is indicated with an asterisk (*). Two asterisks (**) indicate a *p* < 0.01. Results are expressed as mean ± standard deviation (S.D.). A minimum of three biological replicates was used for each experiment.

## Figures and Tables

**Figure 1 genes-15-01647-f001:**
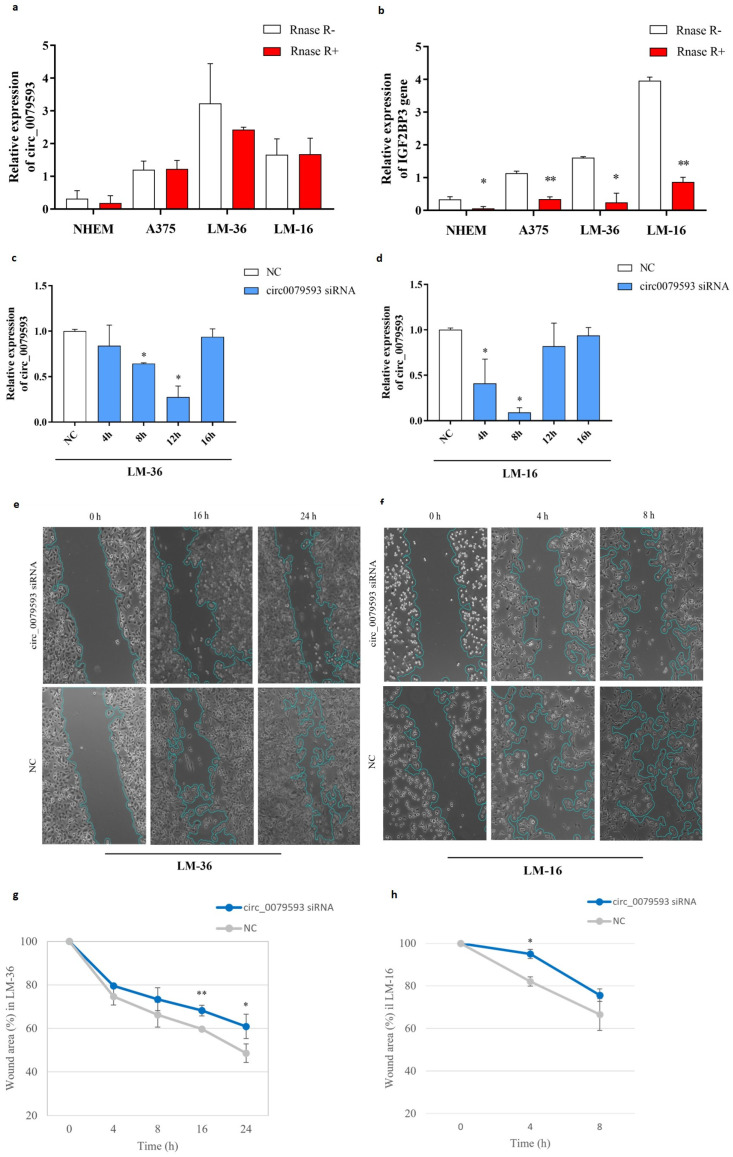
circ_007959 expression in melanoma cell lines and its role in melanoma progression. (**a**) circ_00079593 expression after RNase R treatment in NHEM, A375, LM-36, and LM-16 cell lines. (**b**) IGF2BP3 expression in NHEM, A375, LM-36, and LM-16 cell lines, after RNase R treatment (relative expression on Y axes means relative to TBP expression, used as reference gene in qRT-PCR). circ_0079595 expression after siRNA transfection in LM-36 (**c**) and LM-16 (**d**) melanoma cell lines at different time points. Wound healing assay images after si-circ_00079593 transfection in (**e**) LM-36 and in (**f**) LM-16 (only significative results are shown). Graphic representations of wound healing assay results for (**g**) LM-36 and for (**h**) LM-16. NC = negative control; h = hours. * *p* value < 0.05; ** *p* value < 0.01.

**Figure 2 genes-15-01647-f002:**
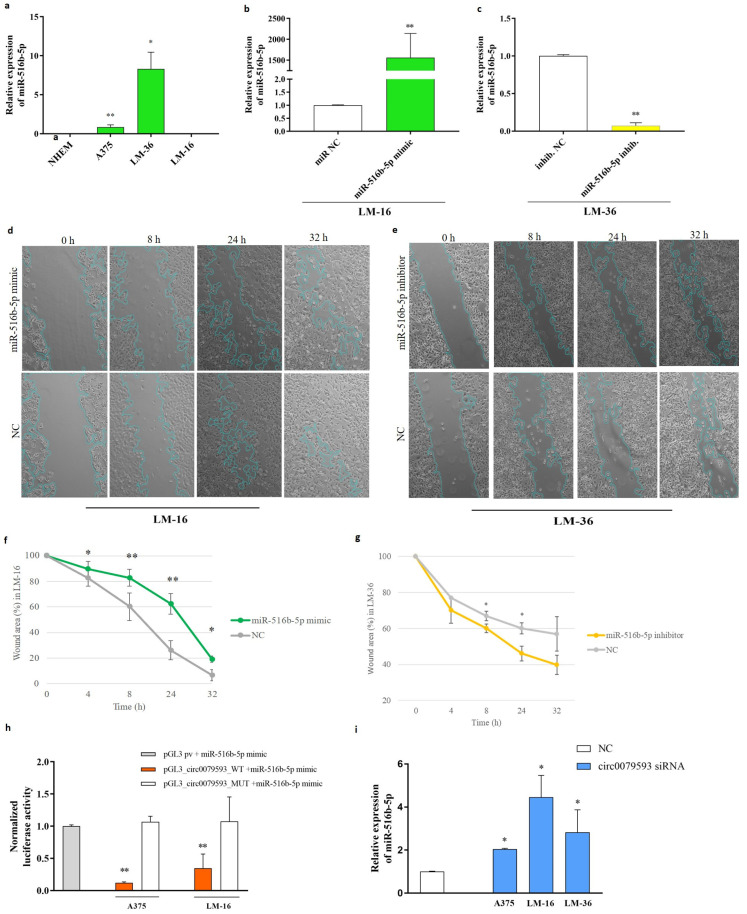
miR-516b-5p expression and its involvement in melanoma metastasis and progression. (**a**) miR-516b-5p expression in NHEM, A375, LM-36, and LM-16 cell lines. (**b**) miR-516b-5p mimic transfection efficiency in LM-36 melanoma cell line. (**c**) miR-516b-5p inhibitor transfection efficiency in LM-16 cells. Wound healing assay images after miR-516b-5p mimic transfection in (**d**) LM-16 and inhibitor transfection in (**e**) LM-36. Graphic representations of wound healing assay results for (**f**) LM-16 and for (**g**) LM-36. (**h**) Dual luciferase reporter assay with wild type and mutated plasmids for miR-516b-5p seed region in circ_0079593 sequence, in A375 and LM-16 melanoma cell lines. (**i**) miR-516b-5p expression after si-circ_0079593 transfection. NC = negative control; h = hours. * *p* value < 0.05; ** *p* value < 0.01.

**Figure 3 genes-15-01647-f003:**
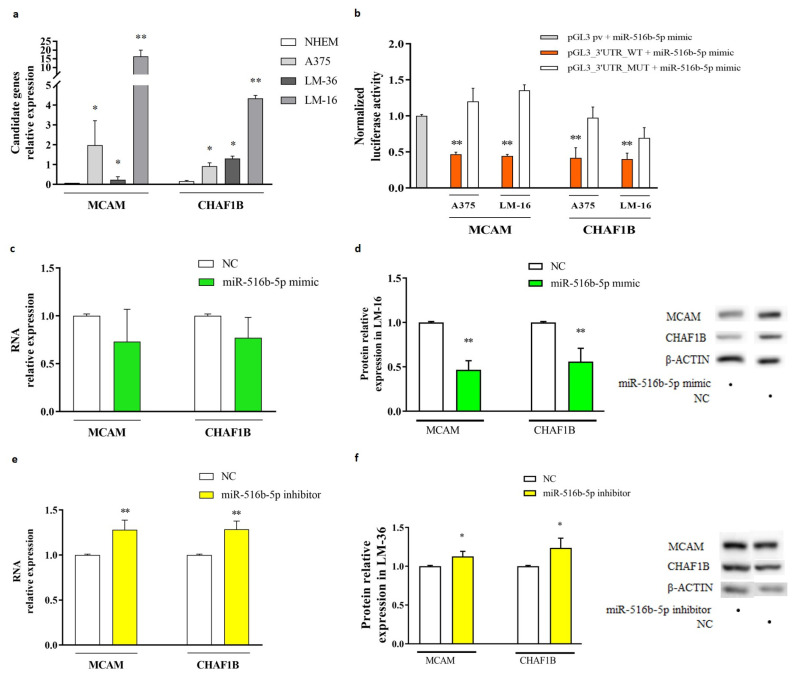
MiR-516b-5p and the interaction with MCAM and CHAF1B targets. (**a**) MCAM and CHAF1B expression in A375, LM-36, and LM-16 respect to NHEM cells. (**b**) Dual luciferase reporter assay with wild type and mutated plasmids for miR-516b-5p seed region in MCAM and CHAF1B 3’UTRs, in A375 and LM-16 melanoma cell lines. (**c**,**d**) Real-time and Western blot analysis of MCAM and CHAF1B genes and proteins with miR-516b-5p mimic transfection in LM-16, and (**e**,**f**) inhibitor transfection in LM-36. NC = negative control; h = hours. * *p* value < 0.05; ** *p* value < 0.01.

**Table 1 genes-15-01647-t001:** Primers used for the analysis of mRNA expression and hsa-circ-0079593.

**Genes**	**Forward and Reverse Primer**	**Product**
CHAF1B	F: 5’-CCTGGAAAAGCCACTCTTGC-3’	139 bp
	R: 5’-ACAGAAGCACGGAATCCTCC-3’	
MCAM	F: 5’-AAGGAGAGGAAGGTGTGGGT-3’	119 bp
	R: 5’-TGGTCTTGTTCACTTGCCGT-3’	
TBP	F: 5’-TGTATCCACAGTGAATCTTGG-3’	102 bp
	R: 5’-ATGATTACCGCAGCAAACC-3’	
IGF2BP3	F: 5’-CCATAGAAGTTGAGCACTCGGTCC-3’	126 bp
	R: 5’-TCTCCACCACTCCATACAGGACTAG-3’	
**circRNA**	**Forward and Reverse Primer**	**Product**
hsa_circ_0079593	F: 5’-AATCTGAACGCCTTGGGTCT-3’	172 bp
	R: 5’-CTCAGCTTTGGCACATGTCT-3’	

**Table 2 genes-15-01647-t002:** List of primers used for the wild-type vector construction and the site-directed mutagenesis reactions. For plasmid construction, the primers include the XbaI restriction site (single underlined) at the ends of both the genomic DNA sequences corresponding to the 3’UTR regions of CHAF1B and MCAM, which contain the binding sites for miR-516b-5p, and at the sequence end of circ_0079593 after RNA enrichment. For mutagenesis, the primers were designed by substituting two nucleotides (double underlined) in the miR-516b-5p binding site, without introducing a new binding site for other miRNAs while introducing a new restriction site for verification.

**Wild-Type Plasmid**	**Primers for Vector Construction**	**Product**
pGL3_3’UTR_CHAF1B_WT	F: 5’-CCTAAGGTTCTAGAGGAGCGGGACACACTGTAAA-3’	404 bp
	R: 5’-ACCTTAGGTCTAGAAAACAAACACAACTACCTTCCAA-3’	
pGL3_3’UTR_MCAM_WT	F: 5’-CCTAAGTTCTAGAAATCTGAACGCCTTGGGTCTG-3’	418 bp
	R: 5’-ACCTTAGGTCTAGACTCAGCTTTGGCACATGTCT-3’	
pGL3_circ_0079593_WT	F: 5’-CCTAAGGTTCTAGAGAGATGGTGGTGGACTGGTC-3’	199 bp
	R: 5’-ACCTTAGGTCTAGAAGTTCCTGGCTTCTGACCAA-3’	
**Mutated Plasmid**	**Primers Used for Mutagenesis Reaction**	
pGL3_3’UTR_CHAF1B_MUT	F: 5’-ATACTGAATACAACAGCATCCATATGACTGGAGAAAAATCAGTACACATGTC-3’	
	R: 5’-GACATGTGTACTGATTTTTCTCCAGTCATATGGATGCTGTTGTATTCAGTAT-3’	
pGL3_3’UTR_MCAM_MUT	F: 5’-CAACACTGCAGCTGCAGCTGGATGCTGCTGGG-3’	
	R: 5’-CCCAGCAGCATCCAGCTGCAGCTGCAGTGTTG-3’	
pGL3_circ0079593_MUT	F: 5’-TGCCTTTAACTGTAATAGTGCGCGCTGGATTATACAGCGTCAATTC-3’	
	R: 5’-GAATTGACGCTGTATAATCCAGCGCGCACTATTACAGTTAAAGGCA-3’	

## Data Availability

Data are contained within the article and Appendix A.

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
