# Peer review of "New Axes of Interaction in Circ_0079593/miR-516b-5p Network in Melanoma Metastasis Cell Lines"

_genes, 2024, doi:10.3390/genes15121647_

Round 1
Reviewer 1 Report
Comments and Suggestions for Authors
Comments:
1. Please explain why chose NHEM, A375, LM16, and LM36 cells for the current study?
2. Figure 1A and 1B: what does "relative expression" mean on Y axis? relative to what?
3. Figure 1h: why stops at 8 hours? not 24 or 32 hours?
4. Figure 1: what is n =? how many times of experiments?
5. Figure 2A: any statistical analysis? any significant difference?
6. Figure 2: what is n =? how many times of experiments?
7. Figure 3A: any statistical analysis? any significant difference?
8. Figure 3: what is n =? how many times of experiments?
9. Figure 3D and 3F: for WB analysis, any data from other cell lines? are they similar in results?
10. Please list all the abbreviations for readers.
Author Response
We would like to thank you for your precious comments. Below, you will find the answers to your questions or comments.
Comment 1
Please explain why chose NHEM, A375, LM16, and LM36 cells for the current study?
Response 1
In this study, the criteria for the choice of the cell lines used were to validate the interaction between circ0079593 and miR516-5p in the different metastatic stages of melanoma. In particular, we found an absence of studies in the literature that used melanoma metastasis-derived cell lines to analyze the circRNA-miRNA-mRNA pattern. This led us to investigate whether there are also effects in the two short-term cell lines derived from melanoma metastases: LM16, derived from lymph node metastasis, and LM36, derived from skin metastasis. These cell lines, in addition to representing advanced stages of tumor progression, were consistent with our hypothesis, showing the coherent expression of key pathway components. Furthermore, being derived from patient biopsies, they provide information on molecular and functional alterations that cannot always be replicated in commercial lines. To ensure comparability with previous studies, however, we paired these lines with a commercial cutaneous melanoma line (A375), widely used in the literature, and a primary melanocyte line (NEHM), used as a control to identify specific alterations associated with melanoma.
Comment 2
Figure 1A and 1B: what does "relative expression" mean on Y axis? relative to what?
Response 2
In Figure 1A and 1B relative expression means relative to TBP gene. It was considered a good reference gene for the qPCR normalization, and we used it to apply the ΔΔCt method. We add in the description of the figure: relative expression on Y axes means relative to TBP expression, used as reference gene in qRT-PCR. It was highlighted in yellow.
Comment 3
Figure 1h: why stops at 8 hours? not 24 or 32 hours?
Response 3
Thank you for your comment. Upon checking, we realized that owing to an error, the scratch test pictures were reversed. Figures 1e and 1f have now been correctly switched.
All scratch test acquisitions were performed at 4, 8, and 24 h, and in some cases, also at 16 and 32 h after the scratch and the respective transfection. In the case of the LM-16 cell line, the acquisitions for figure 1h ended earlier, as the scratch area was no longer detectable at 16 and 24 h. This cell line showed a high migration speed; it was only possible to observe the scratch area 24 and 32 h after transfection with mimic 516b-5, which showed a significant reduction in the speed of scratch healing.
Comment 4
Figure 1: what is n =? how many times of experiments?
Response 4
All scratch test experiments were performed in triplicates on different days. In the figure description we added all the abbreviations present.
Comment 5
Figure 2A: any statistical analysis? any significant difference?
Response 5
Thank you for your comment. We have now replaced the image with the correct one, which includes the significance compared to the NHEM cell line.
Comment 6
Figure 2: what is n =? how many times of experiments?
Response 6
The experiments were performed in triplicates. In the caption, we added all the abbreviations present.
Comment 7
Figure 3A: any statistical analysis? any significant difference?
Response 7
Thank you for your comment. We have now replaced the image with the correct one, which includes the significance in comparison with the NHEM cell line.
Comment 8
Figure 3: what is n =? how many times of experiments?
Response 8
The experiments were performed a minimum of three times. In the figure we added all the abbreviations present.
Comment 9
Figure 3D and 3F: for WB analysis, any data from other cell lines? are they similar in results?
Response 9
We did not do further analysis in other different cell lines.
Comment 10
Please list all the abbreviations for readers.
Response 10
Thank you for your comment. We listed here the abbreviations contained in the article, and in the text, we added the forgotten ones.
NcRNAs Non-coding RNAs
circRNAs/circ CircularRNAs
MiRNAs/miR MicroRNAs
mRNAs Messenger RNAs
CeRNAs Competitive endogenous RNAs
EGFR Epidermal growth factor receptor
ABHD2 Abhydrolase Domain Containing 2, Acylglycerol Lipase
GRM3 Glutamate Metabotropic Receptor 3
MCAM Melanoma Cell Adhesion Molecule
CHAF1B Chromatin Assembly Factor 1 Subunit B
IGF2BP3 Insulin Like Growth Factor 2 MRNA Binding Protein 3
qRT-PCR Quantitative Real-Time PCR
NHEM Normal Human Epidermal Melanocytes
A375 Melanoma cell line CRL-1619, ATCC
LM-16 Melanoma cell line from lymph node metastasis , 4023M
LM-36 Melanoma cell line from skin metastasis, 26414M
SiRNA Silencing RNA
NC Negative control
Reviewer 2 Report
Comments and Suggestions for Authors
This manuscript describes the involvement of the circular RNA, circ_0079593, that can sponge miR-516b-5p, in the regulation of the expression of two melanoma-associated genes encoding the Melanoma Cell Adhesion Molecule (MCMA) and Chromatin Assembly Factor 1 Subunit B (CHAF1B). Thus, the manuscript suggests that circ_0079593 promotes the migration of metastatic melanoma cells, increasing its aggressiveness through the regulation of the miR-516b-5p/MCAM and miR-516b-5p/CHAF1B axes.
Two new distinct types of melanoma metastasis cell lines are used, although a well-established line cutaneous melanoma and a control line of non-malignant melanoma was also used. Methods are correct. Those proteins were identified as the most significantly overexpressed genes in melanoma metastasis cells using several databases that predict specific miRNA-mRNA interactions. Results are clear indicating that there is a link between miR-516b-5p and the two genes in LM-36and consistent with discussion. The manuscript is accompanied with abundant and appropriate supplementary material.
The following minor points would be addressed:
M&M. Please, check the concentration of L-glutamine in the cell culture: 1% L-glutamine (200 nM) seems to be too low.
Right columns at Tables : Products. Are the number referred to bp?. If so, please indicate it.
Author Response
We would like to thank you for your precious comments. Below, you will find the answers to your questions or comments.
Comment 1
M&M. Please, check the concentration of L-glutamine in the cell culture: 1% L-glutamine (200 nM) seems to be too low.
Response 1
Thank you for your comment. Yes, it was wrong. The correct final concentration of glutamine is 2mM. We changed it in the article and highlighted it in yellow.
Comment 2
Right columns at Tables : Products. Are the number referred to bp?. If so, please indicate it.
Response 2
Yes, they are the bp lengths. We add bp to all the numbers, and we highlight them in yellow.
Round 2
Reviewer 1 Report
Comments and Suggestions for Authors
No more comments.